# Factors associated with sexual violence against reproductive-age women in Ghana: A multilevel mixed-effects analysis

Yohannes Mekuria Negussie[1]*, Angwach Abrham Asnake[2], Zufan Alamrie Asmare[3], Bezawit Melak Fente[4], Mamaru Melkam[5], Meklit Melaku Bezie[6], Hiwot Atlaye Asebe[7], Beminate Lemma Seifu[7]

1 Department of Medicine, Adama General Hospital and Medical College, Adama, Ethiopia, 2 Department of Epidemiology and Biostatistics, School of Public Health, College of Medicine and Health Sciences, Wolaita Sodo University, Wolaita Sodo, Ethiopia, 3 Department of Ophthalmology, School of Medicine and Health Science, Debre Tabor University, Debre Tabor, Ethiopia, 4 Department of General Midwifery, School of Midwifery, College of Medicine & Health Sciences, University of Gondar, Gondar, Ethiopia, 5 Department of Psychiatry, College of Medicine and Health Science, University of Gondar, Gondar, Ethiopia, 6 Department of Public Health Officer, Institute of Public Health, College of Medicine and Health Sciences, University of Gondar, Gondar, Ethiopia, 7 Department of Public Health, College of Medicine and Health Sciences, Samara University, Samara, Ethiopia

* yohannes_mekuria@yahoo.com

**Data Availability Statement:** The paper contains all result-based data. In addition, the dataset is

## Abstract

### Background

Sexual violence against women is a pervasive public health challenge and human rights violation, with global prevalence rates of approximately one in three women affected, notably prevalent in African countries. Understanding its multifaceted determinants is crucial for developing targeted intervention strategies. Thus, this study aimed to investigate factors associated with sexual violence against reproductive-age ever-married women in Ghana.

### Methods

A weighted sample of 3,816 reproductive-age ever-married women from the 2022 Ghana Demographic and Health Survey (DHS) was included. To accommodate the hierarchical nature of the DHS data and the binary outcome variable 'sexual violence', a multilevel mixed-effect logistic regression model was employed. The deviance value was utilized for selecting the best-fitted model. In the multivariable multilevel binary logistic regression analysis, adjusted odds ratios (AORs) along with their respective 95% confidence intervals (CIs) were utilized to gauge the association strength, with statistical significance set at a p-value < 0.05.

### Result

The prevalence of sexual violence was found to be 8.80% (95% CI: 7.94–9.74). Factors positively associated with sexual violence included women's decision-making autonomy (AOR = 1.39, 95% CI: 1.08–1.74), husband/partner's alcohol consumption (AOR = 3.88, 95% CI: 2.98–5.06), sex of household head (AOR = 1.31, 95% CI: 1.02–1.68), and justification of

publicly Available and accessible to the MEASURE DHS Program through https://dhsprogram.com/data/dataset_admin/index.cfm.

**Funding:** The author(s) received no specific funding for this work.

**Competing interests:** The authors have declared that no competing interests exist.

**Abbreviations:** AOR, Adjusted Odds Ratio; CI, Confidence Intervals; COR, Crude Odds Ratio; DHS, Demographic and Health Survey; EAs, Enumeration Areas; ICC, Intra-class correlation; MOR, Median Odds Ratio; OR, Odds Ratio.

beating (AOR = 1.35, 95% CI: 1.01–1.81). Conversely, women's age showed a negative association with sexual violence (AOR = 0.68, 95% CI: 0.48–0.98).

## Conclusion

In conclusion, prioritizing initiatives that empower women in decision-making roles, provide support for those struggling with alcohol consumption, and raise awareness about its impact on interpersonal relationships and the risk of sexual violence is essential. Furthermore, addressing harmful gender norms, particularly those justifying violence, and considering demographic characteristics are vital components of comprehensive strategies to prevent and mitigate sexual violence.

## Introduction

Violence against women, including sexual violence, represents a significant public health challenge and a violation of women's fundamental human rights [1]. This violence varies in severity, with sexual violence being one of its most serious forms [2]. Sexual violence encompasses any sexual act or attempt to engage in such acts against someone's consent, including unwanted advances and remarks related to sexuality, regardless of the perpetrator's relationship with the victim or the setting in which it occurs [1, 3]. Sexual violence takes various forms, such as sexual harassment, rape, attempted rape, humiliating sexual acts, inappropriate touching, and unwelcome sexual remarks [3, 4].

Sexual violence has detrimental effects on mothers and their children, affecting physical, mental, and reproductive health [5, 6]. It restricts women's reproductive decision-making autonomy, increasing their vulnerability to sexually transmitted infections, psychological distress, and unwanted pregnancies. Suicidal tendencies, substance abuse, pervasive fear, anxiety, and enduring trauma are also common outcomes [7, 8]. Pregnant women experiencing sexual violence face risks like miscarriages, low birth weight, preterm birth, and intrauterine growth restriction [1, 9–11]. Witnessing violence also increases children's likelihood of perpetrating gender-based violence later on [1].

Globally, about one in three women, or 30%, have faced physical and/or sexual violence [1]. Additionally, it's estimated that 27% of women aged 15–49 years who have been in a relationship have experienced physical, sexual, or both types of intimate partner violence during their lifetime [12]. Approximately 33% of African women have encountered sexual violence at some stage in their lives [13]. In sub-Saharan Africa, around 19% of women have endured sexual violence, with East Africa and West Africa carrying the significant burden of various forms of intimate partner violence [14].

Understanding the multifaceted determinants of sexual violence is crucial for developing targeted intervention strategies. Despite the pervasive nature of sexual violence, it is often underreported due to stigma, fear, and misconceptions, making it a critical area of study. Previous studies revealed that sexual violence is influenced by a multitude of factors spanning individual, familial, communal, and societal realms, which intertwine to either amplify or mitigate the risk. These encompass aspects like place of residence, age, educational attainment, employment status, media exposure, childhood exposure to familial sexual violence, healthcare decision-making, alcohol and drug abuse, socioeconomic status, and prevailing attitudes regarding male entitlement to sexual behavior [1, 7, 15–18].

To effectively address sexual violence, it's crucial to comprehend and mitigate the factors that make individuals more susceptible or resilient to such acts [19, 20]. Despite the

importance of understanding the determinants of sexual violence, there is a paucity of compre-
hensive, nationally representative data on the subject, particularly in countries like Ghana.
Although there have been studies on this issue [21–23], no study has yet investigated the fac-
tors linked to sexual violence in Ghana using the latest data from the Demographic and Health
Survey (DHS) while employing multilevel modeling to address the hierarchical structure
inherent in DHS data [18, 24]. Moreover, many of these studies did not examine community-
level factors that might affect sexual violence and often used small sample sizes. By including
these community-level factors and using larger sample sizes, we can better understand trends
at a national level. Thus, this study aimed to identify factors associated with sexual violence
against reproductive-age Ghanaian women, utilizing recent DHS data. The insights gained will
help create better targeted interventions and policies aimed at combating sexual violence fos-
tering a safer and more equitable society for women. It will also aid in developing and imple-
menting practical solutions to address this important public health issue.

## Methods

### Study design, data source, and participants

The 2022 Ghana DHS is the seventh in a series of DHS surveys in Ghana conducted through
the DHS Program. The survey utilized a cross-sectional study design and gathered data on fun-
damental sociodemographic attributes and various health indicators. The sampling procedure
used in 2022 GDHS is a stratified two-stage cluster sampling by enumeration areas as primary
sampling units and households as secondary sampling units. initially, regions were stratified
into urban and rural areas. In the first stage of selection, target clusters were selected with
probability proportional to size within each stratum. In the second stage of selection, house-
holds were selected within each chosen cluster.

The study included all ever-married women of reproductive age who reported experiencing
sexual violence. Nonetheless, women of reproductive age who had never been married, and/or
had missing data for the sexual violence variable were excluded. For this study, we utilized the
women's dataset (IR dataset), and a weighted sample of 3,816 ever-married reproductive-age
women was included in the final analysis. Details of DHS methodology are available at: https://
dhsprogram.com/Methodology/index.cfm.

### Study variables

**Dependent variable.** The dependent variable was the history of sexual violence perpetrated
by the husband/partner. It was measured based on women's lifetime experiences of one or more
acts of sexual violence perpetrated by their husbands/partners. A participant (woman) who
answered 'yes' to any of the following acts was considered to have experienced sexual violence, and
those who said no to all three questions were considered to have not experienced sexual violence.

1. Physically forced to have sexual intercourse when she did not want to

2. Physically forced to perform any other sexual acts she did not want to

3. Forced with threats or in any other way to perform sexual acts she did not want to

### Independent variables

To align with the study's aims and recognizing the hierarchical structure of the DHS data, the
study considered two levels of independent variables: individual-level factors and community-
level factors.

At level 1 individual characteristics including age, marital status, educational status, employment status, women's decision-making autonomy, husband's/partner's age, husband/partner's working status, husband/partner's educational status, exposure to mass media, husband/partner's alcohol consumption, number of household members, sex of household head, religion, and justification of beating if the wife refuses to have sex with the husband were considered. At level 2, community-level variables such as place of residence, community-level educational status, and community poverty level were examined.

## Operational definitions

**Media exposure.** The study defined exposure to media based on three variables: frequency of radio listening, television watching, and reading newspapers/magazines. Women who engaged in any of these activities at least once a week were categorized as having media exposure, while those who did not were labeled as not having media exposure.

**Community literacy level.** The study looked at the proportion of women in the cluster who attended primary, secondary, and higher education. Combining the educational achievements of individual women in these three categories provided insight into the overall academic profile of the cluster. The group was split based on the national median value, so one category had more educated women and the other had fewer.

**Community poverty level.** The cluster contains a designated percentage of women classified as either poor or poorest. The proportion of poor and poorest women in each cluster was summarized to indicate the overall poverty status within that cluster. Women were sorted based on their poverty level compared to the national median value.

## Data processing and statistical analysis

Data extraction, recoding, and analysis were meticulously executed utilizing STATA version 17. The data were weighted considering sampling weight, primary sampling unit, and strata to guarantee appropriate statistical analysis. In doing so, the representativeness of the survey was restored, and the sampling design was taken into consideration for estimating standard errors.

Descriptive statistics were employed to depict the study population in terms of pertinent attributes. A multilevel binary logistic regression model was employed to accommodate the hierarchical structure inherent in the DHS data. Bivariable multilevel binary logistic regression analysis was fitted to find potential variables for inclusion in the multivariable multilevel binary logistic regression analysis. Variables showing significance with a p-value below 0.20 in this analysis, were considered eligible for further multivariable analysis.

## Model building

The Likelihood Ratio (LR) test, Intra-class Correlation Coefficient (ICC), and Median Odds Ratio (MOR) were computed to assess the degree of heterogeneity among clusters. The ICC assesses the proportion of individual variance in sexual violence among ever-married women to determine the degree of heterogeneity between clusters.

ICC = $\partial^2/ (\partial^2+\pi^2/3)$ [25]. Where: $\partial^2$-between cluster variance, $\pi^2/3$-within-cluster variance

The MOR measures the variation in sexual violence across clusters using the odds ratio scale. It represents the median odds ratio between a cluster with a high likelihood of sexual violence and another with a lower risk when individuals are randomly selected from both clusters or enumeration areas.

MOR = exp $\sqrt{(2*\partial 2*0.6745)}$ ~ MOR = exp $(0.95*\partial)$ [26]. $\partial^2$ indicates that cluster variance

After selecting the variables for multivariable analysis, four models were fitted to determine the best-fitted model. The models were as follows: Model I, or the null model, which is a model with only the outcome variable to determine the extent of cluster variation; Model II, a model with only the individual-level independent variables; Model III, a model with only the community-level independent variables; and Model IV, or the full model, which is a model with both the individual and community-level variables at the same time to determine their effects on the outcome variable. Deviance, or the -2 log-likelihood ratio (-2LLR) statistic, was utilized to evaluate the fitness of the model, with the one displaying the lowest deviance regarded as the best-fitted model. In the final model, the Adjusted Odds Ratio (AOR) and its 95% confidence interval (CI) were used to estimate the strength of the association, and variables with a p-value < 0.05 were considered statistically significant.

## Ethical consideration

Ethical approval and informed consent were not required for this study because it involved a secondary analysis of publicly available survey data. We obtained authorization to download and use the data from http://www.dhsprogram.com specifically for this study.

## Results

### Descriptive characteristics of participants

A total weighted sample of 3816 participants was included in the study, with nearly half of the population aged 35 and above, 52% having a secondary school educational level, and 86% being employed. A total of 86% of participants had mass media exposure; 65% were from households with less than 6 members, and 63% were from male-headed households. Moreover, 54% of women resided in urban areas, while 57% were from communities with a low level of poverty. In this study, the prevalence of sexual violence was 8.80% (95% CI: 7.94–9.74) *(Table 1)*.

### Factors associated with sexual violence

Table 2 presents the adjusted odds ratios (AOR) and their corresponding 95% confidence intervals (CI), demonstrating the relationship between the independent variables and sexual violence. The multilevel multivariable binary logistic regression model identified age, women's decision-making autonomy, husband/partner alcohol consumption, sex of household head, and the justification of beating if a wife refuses to have sex with her husband as factors significantly associated with sexual violence.

The odds of experiencing sexual violence were 33% (AOR = 0.68, 95% CI: 0.48–0.98) lower among women between the ages of 25 and 35 compared to those aged 15 to 24. In comparison to women with decision-making autonomy, those lacking such autonomy experienced a 39% (AOR = 1.39, 95% CI: 1.08–1.74) increase in the odds of sexual violence. Women with husband/partner who drinks alcohol had 3.88 times greater odds of sexual violence as compared to their counterparts. (AOR = 3.88, 95% CI: 2.98–5.06). The odds of sexual violence were 31% (AOR = 1.31, 95% CI: 1.02–1.68) higher among female-headed households compared to male-headed households. Furthermore, women who justified beating if she refuses to have sex with her husband experienced a 35% increase in the odds of sexual violence compared to their counterparts (AOR = 1.35, 95% CI: 1.01–1.81).

### Random effect and model comparison

Random effect analysis within the null model was utilized to explore clustering effects on sexual violence. The findings revealed a notable disparity in sexual violence among clusters

**Table 1. Individual & community-level characteristics of the study participants (n = 3,816).**

| Variable | Weighted frequency (%) | Sexual violence | |
|---|---|---|---|
| | | No (%) | Yes (%) |
| Sexual violence | | 3480 (91.20) | 336 (8.80) |
| **Individual level factors** | | | |
| **Age** | | | |
| 15–24 | 514 (13.46) | 477 (92.80) | 37 (7.20) |
| 25–35 | 1,393 (36.51) | 1,284 (92.18) | 109 (7.82) |
| >35 | 1,909 (50.03) | 1,720 (90.09) | 189 (9.91) |
| **Marital status** | | | |
| Currently in union/living with a man | 3,250 (85.18) | 3,003 (92.40) | 247 (7.60) |
| Formerly in union/living with a man | 566 (14.82) | 478 (84.45) | 88 (15.55) |
| **Educational status** | | | |
| No formal education | 882 (23.10) | 799 (90.71) | 82 (9.29) |
| Primary | 602 (15.79) | 547 (90.89) | 55 (9.11) |
| Secondary | 1,996 (52.31) | 1,811 (90.72) | 185 (9.28) |
| Higher | 336 (8.80) | 322 (95.95) | 14 (4.05) |
| **Employment status** | | | |
| No | 527 (13.82) | 478 (90.63) | 49 (9.37) |
| Yes | 3,289 (86.18) | 3,003 (91.30) | 286 (8.70) |
| **Women's decision-making autonomy** | | | |
| Yes | 1,782 (46.71) | 1,658 (93.02) | 124 (6.98) |
| No | 2,034 (53.29) | 1,823 (89.61) | 211 (10.39) |
| **Husband's/partner's age** | | | |
| <20 | 15 (0.39) | 14 (99.23) | 1 (0.77) |
| 21–25 | 158 (4.15) | 144 (90.87) | 14 (9.13) |
| 26–30 | 394 (10.33) | 364 (92.43) | 30 (7.57) |
| >31 | 3,249 (85.13) | 2,958 (91.04) | 291 (8.96) |
| **Husband's/partner's work status** | | | |
| No | 85 (2.22) | 78 (92.30) | 7 (7.70) |
| Yes | 3,731 (97.78) | 3,402 (91.18) | 329 (8.82) |
| **Husband's educational status (n = 3,249)** | | | |
| No formal education | 726 (22.35) | 674 (92.86) | 52 (7.14) |
| Primary | 296 (9.10) | 274 (92.63) | 22 (7.37) |
| Secondary | 1,745 (53.71) | 1,607 (92.08) | 138 (7.92) |
| Higher | 482 (14.85) | 447 (92.71) | 35 (7.29) |
| **Media exposure** | | | |
| No | 537 (14.07) | 495 (92.11) | 42 (7.89) |
| Yes | 3,279 (85.93) | 2,986 (91.06) | 293 (8.94) |
| **Husband/ partner drinks alcohol** | | | |

(*Continued*)

**Table 1.** (Continued)

| Variable | Weighted frequency (%) | Sexual violence | |
|---|---|---|---|
| | | No (%) | Yes (%) |
| **Sexual violence** | | **3480 (91.20)** | **336 (8.80)** |
| No | 2,627 (68.84) | 2,503 (95.28) | 124 (4.72) |
| Yes | 1,189 (31.16) | 978 (82.21) | 211 (17.79) |
| **Number of Household members** | | | |
| Less than six | 2,472 (64.79) | 2,242 (90.71) | 230 (9.29) |
| Six and above | 1,344 (35.21) | 1,238 (92.12) | 106 (7.88) |
| **Sex of household head** | | | |
| Male | 2,408 (63.10) | 2,220 (92.21) | 188 (7.79) |
| Female | 1,408 (36.90) | 1,260 (89.48) | 148 (10.52) |
| **Religion** | | | |
| No religion | 76 (2.00) | 68 (89.41) | 8 (10.59) |
| Catholic | 307 (8.04) | 287 (93.57) | 20 (6.43) |
| Pentecostal/charismatic | 1,564 (40.99) | 1,406 (89.91) | 158 (10.09) |
| Other Christian | 545 (14.28) | 484 (88.87) | 61 (11.13) |
| Islam | 782 (20.48) | 743 (95.10) | 39 (4.90) |
| Other | 542 (14.21) | 491 (90.60) | 51 (9.40) |
| **Beating justified if the wife refuses to have sex with the husband** | | | |
| No | 3,290 (86.22) | 3,006 (91.38) | 284 (8.62) |
| Yes | 526 (13.78) | 474 (90.12) | 52 (9.88) |
| **Community level factors** | | | |
| **Place of residence** | | | |
| Urban | 2,043 (53.53) | 1,846 (90.35) | 197 (9.65) |
| Rural | 1,773 (46.47) | 1,635 (92.20) | 138 (7.80) |
| **Community-level women education** | | | |
| High education level | 2,179 (57.10) | 1,983 (90.98) | 196 (9.02) |
| Low education level | 1,637 (42.90) | 1,498 (91.50) | 139 (8.50) |
| **Community-level poverty level** | | | |
| Low poverty level | 2,180 (57.13) | 1,979 (90.78) | 201 (9.22) |
| High poverty level | 1,636 (42.87) | 1,501 (91.77) | 135 (8.23) |

(ICC = 13.25%), suggesting that these clusters explain 13.25% of the variance in sexual violence. To identify the factors associated with sexual violence, Model III encompassing both individual and community-level variables was chosen as the best-fitted model, given its lowest deviance compared to other models (2,265.94) (*Table 2*).

**Table 2. Multilevel analysis of factors associated with sexual violence against ever-married reproductive-age Ghanaian women (n = 3,816).**

| Variable | Null model | Model I | Model II | Model III |
|---|---|---|---|---|
| | | AOR (95%CI) | AOR (95%CI) | AOR (95%CI) |
| **Individual level characteristics** | | | | |
| **Age** | | | | |
| 15–24 | | 1 | | 1 |
| 25–35 | | 0.71 (0.49–1.01) | | 0.67 (0.48–0.98) * |
| >35 | | 0.81 (0.57–1.16) | | 0.77 (0.61–1.18) |
| **Educational status** | | | | |
| No formal education | | 1 | | 1 |
| Primary | | 0.95 (0.66–1.37) | | 0.91 (0.63–1.34) |
| Secondary | | 0.98 (0.72–1.34) | | 0.92 (0.65–1.29) |
| Higher | | 0.63 (0.35–1.14) | | 0.56 (0.31–1.04) |
| **Employment status** | | | | |
| No | | 1 | | 1 |
| Yes | | 0.84 (0.61–1.16) | | 0.85 (0.61–1.18) |
| **Woman's decision-making autonomy** | | | | |
| Yes | | 1 | | 1 |
| No | | 1.36 (1.06–1.74) | | 1.39 (1.08–1.74) * |
| **Media exposure** | | | | |
| No | | 1 | | 1 |
| Yes | | 1.23 (0.88–1.72) | | 1.15 (0.82–1.62) |
| **Husband/ partner drinks alcohol** | | | | |
| No | | 1 | | 1 |
| Yes | | 3.87 (2.96–5.05) | | 3.88 (2.98–5.06) * |
| **Number of Household members** | | | | |
| Less than six | | 1 | | 1 |
| Six and above | | 1.02 (0.80–1.33) | | 1.05 (0.83–1.33) |
| **Sex of household head** | | | | |
| Male | | 1 | | 1 |
| Female | | 1.33 (1.04–1.72) | | 1.31 (1.02–1.68) * |
| **Religion** | | | | |
| Pentecostal/charismatic | | 1 | | 1 |
| No religion | | 1.57 (0.83–2.97) | | 1.58 (0.84–3.00) |
| Catholic | | 0.69 (0.45–1.06) | | 0.71 (0.46–1.09) |
| Other Christian | | 0.89 (0.61–1.30) | | 0.89 (0.61–1.30) |
| Islam | | 1.08 (0.76–1.55) | | 1.05 (0.73–1.51) |
| Other | | 0.99 (0.68–1.45) | | 1.00 (0.67–1.45) |
| **Beating justified if a wife refuses to have sex with the husband** | | | | |
| No | | 1 | | 1 |
| Yes | | 1.34 (0.99–1.80) | | 1.35 (1.01–1.81) * |
| **Community level characteristics** | | | | |
| **Place of residence** | | | | |
| Urban | | | 1 | 1 |
| Rural | | | 0.87 (0.67–1.13) | 0.75 (0.54–1.03) |
| **Community-level women's education** | | | | |
| High education level | | | 1 | 1 |
| Low education level | | | 0.95 (0.74–1.22) | 1.01 (0.73–1.39) |
| **Community-level poverty level** | | | | |

*(Continued)*

**Table 2.** (Continued)

| Variable | Null model | Model I | Model II | Model III |
|---|---|---|---|---|
| | | AOR (95%CI) | AOR (95%CI) | AOR (95%CI) |
| Low poverty level | | | 1 | 1 |
| High poverty level | | | 0.92 (0.69–1.22) | 0.95 (0.67–1.35) |
| **Random effect analysis** | | | | |
| ICC | 13.25% | 11.53% | 12.71% | 10.63% |
| log-likelihood | -1212.79 | -1135.57 | -1211.12 | -1132.97 |
| Deviance | 2,425.58 | 2,271.14 | 2,422.24 | 2,265.94 |
| AIC | 2429.58 | 2309.15 | 2432.24 | 2309.93 |
| BIC | 2442.28 | 2429.78 | 2463.99 | 2449.61 |

**Notes:** *significant at p<0.05 in adjusted regression analysis, 1 = Reference

**Abbreviations:** AIC: Akaike information criteria; AOR: Adjusted odds ratio; BIC: Bayesian information criteria; CI: Confidence interval; ICC: Intra-class correlation coefficient

## Discussion

This study aimed to assess the factors associated with sexual violence against reproductive-age ever-married women in Ghana. The prevalence of sexual violence was found to be 8.80%. In the multivariable mixed-effect binary logistic regression model, factors such as being aged 25–35 years, lack of women's decision-making autonomy, husband or partner alcohol consumption, belonging to a female-headed household, and justification of beating if a wife refuses to have sex with her husband were identified as significant factors associated with sexual violence.

The study unveiled a prevalence of sexual violence of 8.80% (95% CI: 7.94–9.74). This finding is congruent with studies conducted in Zimbabwe (8.9%) [27] and in 26 sub-Saharan African countries in 2020 (8.7%) [28]. Conversely, it was lower than studies done in Haiti (10.5%) [29], Rwanda (12%) [16], Kenya (16%) [30], Uganda (24.3%) [7], and a systematic review and meta-analysis of 9 cross-sectional studies in Africa (33%) [13]. Moreover, the prevalence of sexual violence in the current study was higher than in studies done in Liberia (6.56%) [31], Myanmar (3.8%) [32], and Nigeria (3%) [33]. The discrepancies in the prevalence of sexual violence across different studies and countries can be ascribed to methodological disparities, divergent definitions of sexual violence, cultural and societal distinctions, challenges related to reporting and disclosure, timing discrepancies in data collection, and variations in population characteristics.

This study revealed that the odds of experiencing sexual violence were lower among women between the ages of 25 and 35 years compared to those aged between 15 and 24 years. This finding is similar to studies [34–36]. This may be because of increased awareness, greater life experience, larger social circles, improved access to education and employment, and potentially more stable and supportive relationships, all of which may contribute to a decreased vulnerability to such violence among older women. Conversely, various studies have produced inconsistent findings about the relationship between sexual violence and age [37, 38]. The relationship between age and sexual violence is complex, necessitating further studies to elucidate a definitive association.

In agreement with previous studies conducted in Rwanda [16], Uganda [7], India [39], and Kenya [30], the current study found that women lacking decision-making autonomy had increased odds of experiencing sexual violence. This could be because of the restricted ability of women to make decisions about their own lives, particularly in matters related to sexuality, finances, and household affairs, which renders them more vulnerable to coercion, manipulation, and exploitation, heightening their risk of encountering sexual violence perpetrated by

individuals taking advantage of their diminished agency and power within societal structures. Conversely, Women empowered by decision-making autonomy have the agency to safeguard their rights, boasting higher self-esteem and control over their lives, which diminishes their vulnerability to sexual violence [40–42].

The odds of sexual violence were higher among women with a husband/partner who drinks alcohol compared to those with a husband/partner who doesn't drink alcohol. This finding is supported by previous studies; [7, 16, 39, 43]. The justification could be alcohol can impair judgment and lower inhibitions, leading to situations where individuals may engage in harmful behavior such as sexual assault. Moreover, alcohol misuse may exacerbate existing relationship conflicts and contribute to power imbalances within partnerships, further increasing the likelihood of sexual violence.

In this study, compared to women in male-headed households, those in female-headed households had greater odds of experiencing sexual violence. This finding is consistent with a study conducted in Rwanda, which reported that women from male-headed households had 48% lower odds of experiencing sexual violence [16]. The justification could be the lack of a male protector in the household may make women more susceptible to external threats and violence. Moreover, households led by women may face increased risks, as cultural norms of female submissiveness may lead to assertiveness, resulting in relational tensions that elevate the risk of sexual violence. Conversely, households led by women may face increased risks, as cultural norms of female submissiveness may lead to assertiveness, resulting in relational tensions that elevate the risk of sexual violence [44].

Beating justified if a wife refuses to have sex with the husband was another significant factor associated with sexual violence. Women who justified beating in such circumstances experienced a 35% increase in the odds of sexual violence compared to their counterparts. This is in line with studies undertaken in Rwanda [16], Uganda [7, 45], Nigeria [46], and Myanmar [32] and a previous study in Ghana [18]. This could be attributed to patriarchal norms that endorse male dominance and control, fostering violence against women. Furthermore, women may prioritize preserving the family's reputation over addressing their health issues, which exacerbates the problem. Moreover, in patriarchal societies, women often accept male supremacy and view the husband's right to use violence against his wife as a punitive measure [47, 48].

## Strengths and limitations of the study

The study relied on the DHS dataset, sourced from standardized procedures for data collection, thereby bolstering the validity of its findings. By employing weighted data, the study ensured that its findings accurately represented the national-level demographics. Additionally, the study's use of the latest available DHS data contributed to the timeliness and relevance of its findings. However, this research has certain constraints. Its cross-sectional design means it couldn't determine causal relationships. Also, respondents had to rely on their recall, introducing the possibility of recall bias as they recounted past events. Moreover, there's a possibility of social desirability bias during data collection, particularly in sensitive topics like sexual violence, which could affect how respondents reported their experiences. This bias may have influenced the accuracy of estimating the prevalence of sexual violence. However, despite these constraints, the study still provides valuable insights into potential predictors of sexual violence among women in Ghana.

## Conclusion

In this study, lack of decision-making autonomy for women, husband/partner alcohol consumption, belonging to a female-headed household, and the justification of beating if a wife

refuses to have sex with her husband were identified as factors significantly associated with sexual violence.

It is crucial to focus on initiatives that aim to empower women, particularly in decision-making processes within households and communities. Additionally, offering support services for individuals struggling with alcohol consumption issues, including counseling and rehabilitation programs, is essential. Furthermore, raising awareness about the impact of alcohol abuse on interpersonal relationships and the potential for increased risk of sexual violence is imperative. Challenging harmful gender norms perpetuated by justifying beating as a response to a wife's refusal of sex and considering demographic characteristics such as age are also important. Implementing comprehensive strategies and programs to prevent and mitigate sexual violence is crucial for fostering safer and more equitable communities for all.

## Acknowledgments

The authors express their gratitude to the Measure DHS program for providing on-request open access to its dataset.

## Author Contributions

**Conceptualization:** Yohannes Mekuria Negussie, Beminate Lemma Seifu.

**Data curation:** Yohannes Mekuria Negussie, Beminate Lemma Seifu.

**Formal analysis:** Yohannes Mekuria Negussie, Beminate Lemma Seifu.

**Funding acquisition:** Yohannes Mekuria Negussie, Angwach Abrham Asnake, Zufan Alamrie Asmare, Bezawit Melak Fente, Mamaru Melkam, Meklit Melaku Bezie, Hiwot Atlaye Asebe, Beminate Lemma Seifu.

**Investigation:** Yohannes Mekuria Negussie, Beminate Lemma Seifu.

**Methodology:** Yohannes Mekuria Negussie, Beminate Lemma Seifu.

**Project administration:** Yohannes Mekuria Negussie, Angwach Abrham Asnake, Zufan Alamrie Asmare, Bezawit Melak Fente, Mamaru Melkam, Meklit Melaku Bezie, Hiwot Atlaye Asebe, Beminate Lemma Seifu.

**Resources:** Yohannes Mekuria Negussie, Angwach Abrham Asnake, Zufan Alamrie Asmare, Bezawit Melak Fente, Mamaru Melkam, Meklit Melaku Bezie, Hiwot Atlaye Asebe, Beminate Lemma Seifu.

**Software:** Yohannes Mekuria Negussie, Beminate Lemma Seifu.

**Supervision:** Yohannes Mekuria Negussie, Angwach Abrham Asnake, Zufan Alamrie Asmare, Bezawit Melak Fente, Mamaru Melkam, Meklit Melaku Bezie, Hiwot Atlaye Asebe, Beminate Lemma Seifu.

**Validation:** Yohannes Mekuria Negussie, Angwach Abrham Asnake, Zufan Alamrie Asmare, Bezawit Melak Fente, Mamaru Melkam, Meklit Melaku Bezie, Hiwot Atlaye Asebe, Beminate Lemma Seifu.

**Visualization:** Yohannes Mekuria Negussie, Angwach Abrham Asnake, Zufan Alamrie Asmare, Bezawit Melak Fente, Mamaru Melkam, Meklit Melaku Bezie, Hiwot Atlaye Asebe, Beminate Lemma Seifu.

**Writing – original draft:** Yohannes Mekuria Negussie, Beminate Lemma Seifu.

**Writing – review & editing:** Yohannes Mekuria Negussie, Angwach Abrham Asnake, Zufan Alamrie Asmare, Bezawit Melak Fente, Mamaru Melkam, Meklit Melaku Bezie, Hiwot Atlaye Asebe, Beminate Lemma Seifu.

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
