## [Decision Letter · Decision Letter 0]

7 Jul 2024

PONE-D-24-14631Factors associated with sexual violence against reproductive-age women in Ghana: a multilevel mixed-effects analysisPLOS ONE

Dear Dr. Mekuria Negussie,

Thank you for submitting your manuscript to PLOS ONE. After careful consideration, we feel that it has merit but does not fully meet PLOS ONE’s publication criteria as it currently stands. Therefore, we invite you to submit a revised version of the manuscript that addresses the points raised during the review process.

In particular, the referees highlight several common threads.  It is important for you to clarify how your paper relates to and builds on other closely related papers analyzing IPV in Ghana (even if using different data sources).  There are also a number of comments suggesting that you better clarify the construction and selection of variables, and justify the structural of the statistical model.  Please note that four referees provided comments but one was identified as "reviewer zero" by the management system.

We look forward to receiving your revised manuscript.

Kind regards,

Jessica Leight, PhD

Academic Editor

PLOS ONE

Journal Requirements:

Reviewers' comments:

Reviewer's Responses to Questions

**Comments to the Author**

1. Is the manuscript technically sound, and do the data support the conclusions?

Reviewer #1: Partly

Reviewer #2: No

Reviewer #3: Partly

2. Has the statistical analysis been performed appropriately and rigorously? 

Reviewer #1: No

Reviewer #2: No

Reviewer #3: I Don't Know

3. Have the authors made all data underlying the findings in their manuscript fully available?

Reviewer #1: Yes

Reviewer #2: Yes

Reviewer #3: Yes

4. Is the manuscript presented in an intelligible fashion and written in standard English?

Reviewer #1: Yes

Reviewer #2: Yes

Reviewer #3: Yes

5. Review Comments to the Author

Reviewer #1: This paper presents a statistical analysis of recently released data on sexual violence from the Demographic and Health Survey (DHS) in Ghana. The authors use data on individual, partner, household, and community-level characteristics to examine their associations with incidence of sexual violence among married/co-habiting female respondents in Ghana. The paper is timely in providing this evidence since the data has just been released, but it could use some improvements to help address some methodological as well as conceptual issues. In particular, it could be better situated within the Ghanian context of previous studies exploring any kind of violence faced by women. Overall, I found the writing to be well structured and coherent, but lacking in some key areas for clarity. Attached are some suggestions that could help the authors in revising the paper.

Reviewer #2: This paper addresses the main factors explaining the incidence of sexual violence against reproductive-aged women in Ghana, using a 2022 weighted sample of 3,816 reproductive-age women interviewed for the Ghana Demographic and Health Survey. The authors present results on the main determinants such as husband/partner alcohol consumption or women decision-making autonomy. The authors compare their results with the evidence from other African and developing countries.

The topic is certainly of great interest, especially for the consequences on the well-being of an important population subgroup (i.e. women in reproductive age) but also it has relevance for the well-being of minors, and it has spillovers on society overall.

Yet, it appears that the current approach chosen by the authors is a little simplistic on different elements. I list here a few not necessarily in order of importance.

1) It is well known that women who suffered intimate partner violence (IPV), and sexual violence, do not necessarily reveal it in the contest of survey in which the question on sexual abuses is asked directly. It would be important to know more about the way in which the question was used and the method of data collection (direct interviews? Anonymous questionnaires?) Is there any administrative data which allows to corroborate the results of the survey (showing, as expected, that the administrative data underestimate the phenomenon respect to the survey data?). How the author think to address the problem of underreporting when sexual violence is at stake? How does this problem affect their results?

2) The authors focus on IPV from husband or partner. What about ex-husbands or ex-partners? Why is this not included?

3) The authors should better justify the reason for selecting only women in reproductive age. Do they think the health costs are higher? How so?

4) In general, important variables for the authors do deserve a better explanation:

a. It is not clear what alcohol consumption means. I might be someone who consume alcohol, but this does not necessarily mean that I have substance abuse problems.

b. Media exposure for respondents: being exposed less than once a week and at least weekly it is not the same. What is the distribution of this variable?

5) It would be extremely useful to show the results of a basic regression model

Reviewer #3: This paper examines the various factors that influence sexual violence against reproductive age women in Ghana. The authors attempt to identify key factors that may contribute towards women being vulnerable towards sexual violence. I believe the findings from this study can inform policy discussions and interventions to reduce IPV.

I have a few comments that I hope will be useful to the authors.

1) Since the authors are interested in the vulnerabilities of reproductive age women, one of the factors that they should consider in their variables is whether the woman has completed her fertility or not. There is research that shows that this is an important aspect to include because women of reproductive age, and who are still continuing their fertility may not be subject to IPV in the same way as someone who has completed their fertility. Including this variable could also give the readers a better insight as to why there is a negative correlation between age and IPV - especially for the age group 25-35.

2) The result for the sex of the head of the household is not very clear and could benefit from further discussion. The sexual violence variable measures violence perpetrated by the woman's husband/partner. The explanation that this may be lower in male headed households because males can protect doesn't seem very plausible.

3) The positive correlation between the attitude towards beautiful if the wife refuses sex with the husband and sexual violence seems obvious. I wonder if that shouldn't be considered as a factor to explore since it is almost a given in the end.

4) The authors could also explore differences in levels of education and age gaps between spouses to see if these play a larger role in the prevalence of IPV.

5) Typo on page 7 - it should be 3816 respondents instead of 3186

6) Lastly, the authors can look into some more studies that have in fact looked at the factors that contribute towards sexual violence against this cohort of women. While these studies may not have used the latest rounds of the Ghananian DHS, they have in fact looked into similar questions. The authors can use these to motivate their own work, and add as to how their work is complementing prior work. I add a few citations below:

a) Asiedu, Christobel. "Lineage ties and domestic violence in Ghana: Evidence from the 2008 demographic and health survey." Journal of Family Issues 37.16 (2016): 2351-2367.

b) Tenkorang, Eric Y., et al. "Factors influencing domestic and marital violence against women in Ghana." Journal of family violence 28 (2013): 771-781.

6. PLOS authors have the option to publish the peer review history of their article (what does this mean?). If published, this will include your full peer review and any attached files.

Reviewer #1: **Yes: **Anirudh Tagat

Reviewer #2: No

Reviewer #3: No

---

## [Author Response · Author response to Decision Letter 0]

3 Aug 2024

Reviewer null

Reviewer Concern 1: Sampling Procedure:

1) You employed a stratified two-stage cluster sampling design.

First Stage: Target Clusters Selection

o Target clusters were selected from the sampling frame using probability proportional to size.

o Clarification on Cluster Units: The unit of cluster refers to the residency type, such as urban or rural areas. Stratification was based on these conditions to ensure representation across different residency types.

Second Stage: Household Selection

o In the second stage, households (HH) were selected within each chosen cluster.

Author’s response: Dear reviewer, thank you very much for your valuable comment and suggestion to improve the quality of the manuscript. We have updated the sampling procedure section as per your suggestion. (See the revised manuscript, Methods section)

Reviewer concern 2. Variables:

Dependent Variable: Sexual violence in a lifetime.

o Recommendation: Consider using a more recent outcome rather than a lifetime measure. Lifetime measures may encompass a range of determinants and outcomes, making it difficult to isolate specific factors. This should be noted as a limitation.

Author’s response: Dear reviewer, thank you for your valuable insights. Our study conducted a secondary analysis of DHS data, using the information directly from the dataset. We agree that lifetime measures can cover a broad spectrum of determinants and outcomes, making it challenging to isolate specific factors.

Reviewer concern 3. Analysis Section:

The data were weighted to account for the sampling weight, primary sampling unit, and strata, ensuring appropriate statistical analysis.

o Primary Sampling Unit and Strata Variables: It's beneficial to specify the variables used for the primary sampling unit, strata, and weighting to educate the reader on the methodology.

Author’s response: Dear reviewer, thank you for your valuable insights. The variables used for primary sampling unit, stratification used in the design and weighting were v021, v023 and v005, respectively. 

Reviewer concern 4. Model Building:

Formulas for the Intraclass Correlation Coefficient (ICC) and Median Odds Ratio (MOR) were used but not explained.

o Symbols Explanation: Please provide descriptions for each symbol used in the formulas.

Author’s response: Dear reviewer, thank you very much for your professional insight. We have corrected it. (See the revised manuscript, Methods section)

Reviewer concern 5. Result session:

Random Effect and Model Comparison:

o Placement of Description: It's recommended to place the description of random effects and model comparisons after assessing the factors associated with sexual violence. The model conclusions on the variance effects related to clustering should follow the findings of the association.

Author’s response: Dear reviewer, thank you very much for your recommendation. We have made corrections as per your recommendation. (See the revised manuscript, Result section)

Reviewer concern 6. Discussion:

o Summary of Findings: Start by summarizing the key findings of your analysis before discussing them in detail.

o Comparison with Other Countries: While comparing the prevalence of sexual violence with other African countries, include specific figures from Ghana for a comprehensive comparison.

o Time Component in Prevalence: Clarify whether the prevalence is for a lifetime or a 12-month period. Typically, prevalence is a rate with a time component, so it's important to specify this.

The second paragraph in the discussion and the first paragraph in the conclusion should be revised for clarity:

"In the multivariable mixed-effect binary logistic regression model, variables such as age, women’s decision-making autonomy, husband/partner alcohol consumption, sex of household head, and justification of beating if a wife refuses to have sex with her husband were found to be significant factors associated with sexual violence."

o Clarification of Variables: Provide specific directions for each variable:

Age: Specify the age groups associated with increased risk.

Women’s decision-making autonomy: Indicate women with no decision-making autonomy as being more at risk.

Husband/partner alcohol consumption: State that alcohol consumption by the husband/partner increases the risk.

Sex of household head: Specify if the sex of the household head (e.g., male-headed households) is a factor.

Justification of beating: Indicate that the belief in the justification of beating if a wife refuses sex is associated with higher risk.

Author’s response: Dear Reviewer, we are grateful for your constructive feedback and helpful suggestions, which have enabled us to enhance this manuscript significantly. We have made amendments incorporating all your comments and suggestions. (See the revised manuscript, discussion section)

Reviewer #1

Summary

This paper presents a statistical analysis of recently released data on sexual violence from the Demographic and Health Survey (DHS) in Ghana. The authors use data on individual, partner, household, and community-level characteristics to examine their associations with incidence of sexual violence among married/co-habiting female respondents in Ghana. The paper is timely in providing this evidence since the data has just been released, but it could use some improvements to help address some methodological as well as conceptual issues. In particular, it could be better situated within the Ghanian context of previous studies exploring any kind of violence faced by women. Overall, I found the writing to be well structured and coherent, but lacking in some key areas for clarity. Below are some suggestions that could help the authors in revising the paper.

Author’s response: Dear Reviewer, we are grateful for your constructive feedback and helpful suggestions, which have enabled us to enhance this manuscript significantly. We appreciate your commitment to excellence and scholarly dialogue.

Major Comments:

Reviewer concern: Motivation for the paper: The paper starts off well by describing the importance of studying sexual violence. However, it could do with more emphasis on why authors are focusing on sexual violence over other types of violence faced by women (and on which data is available as well). This would help strengthen the argument and also streamline later parts of the discussion.

Author’s response: Dear reviewer thank you for your professional insight. We have carefully revised the introduction section in response to your valuable comments and suggestions. We appreciate your guidance in helping us enhance our work. (See the revised manuscript, Introduction section)

Reviewer concern: Contribution to literature: As mentioned above, we don’t know enough about how this study differs from (or how it adds to) existing literature on sexual (and other types of) violence in Ghana. Presumably, the DHS has previous editions in Ghana, and it would be good to cite this evidence as a starting point. On p.4, line 90, the authors argue for the novelty of the study, but do not speak more to this point. Although I am not an expert on studies of this nature in Ghana, I was able to find a few studies that indeed do contribute to this. For example, Tenkorang et al. 2013 have done this with the 2012 data from DHS Ghana. There is also qualitative work on this domain that could help sharpen the discussion (Apatinga and Tenkorang, 2021). More on this below.

Author’s response: Dear reviewer thank you for your valuable comment and professional insight. Our study includes new variables, different analytical approaches, and more recent data that reflect changes in societal dynamics since previous studies. This allows us to capture current trends and changes in the patterns of sexual violence. While previous studies, such as Tenkorang et al. (2013) using 2012 DHS data, have laid the groundwork for understanding these issues, our study introduces new variables and employs advanced analytical techniques to provide a more nuanced analysis. Additionally, qualitative insights from works like Apatinga and Tenkorang (2021) enrich our understanding of the sociocultural contexts of violence, allowing for a more comprehensive discussion. These elements together underscore the unique contribution of our research to the existing body of literature. (See the revised manuscript, lines 90-104, introduction section)

Reviewer concern: Variables: The variable that is the key dependent variable is not defined very clearly. On p.5, line 116, the authors define it as a percentage, but this would only be true if the estimation was run at a community level. Since the authors are making use of multi-level logistic regression models, my assumption is that this is likely an error in definition. Later on (line 120) they discuss how the main dependent variable of sexual violence takes a value of 1 if certain criteria are met and 0 otherwise, suggesting that it is indeed not measured in percentage terms. Given that the authors use three variables as conditions to determine whether or not the respondent experienced sexual violence, it might also be worth exploring – as a sensitivity check – what the results look like if they were to make the criteria more or less stringent, or even just using one question as a measure of experience of sexual violence.

Author’s response: Dear reviewer Thank you for your critical comment. All the variables, including the dependent variable, were extracted from the DHS dataset, and the definitions of the dependent variables were derived directly from the DHS. Based on your comment, we have made modifications to the manuscript to ensure clarity and avoid any misinterpretation or ambiguity. (See the revised manuscript, lines 122-1230, Methods section)

Reviewer concern: Community-level variables: The authors include as part of their multi-level estimates community variables that capture community-level poverty and literacy. First, there is no mention of these beforehand coming from the literature or else- where, so it would be good to set up the motivation for why we think community- level is important to explore. Second, it also becomes important to then account for secondary data or any other policy-type variables such as laws related to sexual violence (Bhuwania et al., 2024). One last note on using community-level variables, or aggregating using household or respondent level data – it would be good to have information on how exactly this was done. For example, are these weighted aver- ages, do they include only women who have faced violence, or all women, or all individuals/households? More importantly, is it net-of-own aggregate, so that it excludes the current woman/household from the estimation? These are important methodological concerns that need to be detailed, even if only in a footnote.

Author’s response: Dear reviewer thank you for your comments and questions. The motivation for including community level variables is the assumption High levels of poverty can increase stress and frustration, which may contribute to higher rates of violence, including sexual violence. Literacy and education levels can influence awareness and attitudes towards sexual violence. Communities with higher literacy rates may have better access to information about rights, resources, and support systems. Variables which are related with laws regarding sexual violence were not available in the GDHS due to this reason we couldn’t include it in the analysis. How the community level variables were aggregated is stated as follows 

Community literacy level: The study looked at the proportion of women in the cluster who attended primary, secondary, and higher education. Combining the educational achievements of individual women in these three categories provided insight into the overall academic profile of the cluster. The group was split based on the national median value, so one category had more educated women and the other had fewer.

Community poverty level: The cluster contains a designated percentage of women classified as either poor or poorest. The proportion of poor and poorest women in each cluster was summarized to indicate the overall poverty status within that cluster. Women were sorted based on their poverty level compared to the national median value.

Reviewer concern: Statistical model: The model as it stands is not sufficiently motivated, and by p.6 when a reader lands on it, it is unclear what the basis for this model is. It is also unclear why authors use a non-standard level of statistical significance (20%) to consider variables for further analysis. Again, calling back to the literature on using multi-level mixed effects model to study similar questions would be useful. Further, it might be good to have some info on what the benefit of selecting these models in this step-wise way. What is the benefit of choosing pre-determined variables over, say, using a lasso model or similar to find covariates that maximize the fit (if that is the goal of the model).

Author’s response: Dear reviewer thank you for your comments and questions. Even though we stated that we used a bivariable analysis for variable selection we have included all the potential predictors on the multivariable analysis except those variables which has a smaller observation than the total sample size. The purpose of fitting multiple models in step-wise way were, as you aknow DHS data has a hierarchical nature, which means variables are measured at different levels like individual level and community level. We fit a null model without explanatory variables to determine the extent of cluster variation in healthcare-seeking behavior. We have also fitted a model with individual-level variables only, community-level variables only, and both individual and community-level variables to see which model will explain our data better. This method employed to compare the models and identify which level of variables (individual, community, or both) provides the best explanation for sexual violence. It’s a robust way to understand the data’s structure and the factors influencing the outcome.

Reviewer concern: Discussion: As with the introduction, the discussion section needs more work to better situate the findings of the model with other studies from Ghana. It would also be good, where appropriate, to mention findings from qualitative studies as well (Cannon et al., 2020; Apatinga and Tenkorang, 2022). This might help bridge some gaps in terms of what the authors find and are able to link to Ghanian context. This is also a good place to talk about any cultural beliefs or social norms around sexual violence in Ghana, since the prevalence is also being discussed. The authors could use some sub-group analysis (instead of multi-level models) to understand how poverty and literacy could vary the levels of sexual violence and their determinants. This is an important part of the story that needs to be included carefully.

Author’s response: Dear reviewer, thank you very much for your constructive feedback and professional suggestions to enhance the quality of the manuscript. We carefully developed the discussion section to align closely with our findings and selected references from a comprehensive range of relevant literature to support our study's results and research question. While the studies you suggested are intriguing and methodologically sound, we are concerned that their findings might differ from ours and may not directly address our study’s research questions. Hence, we believe it is necessary to reconsider the inclusion of some of the suggested studies due to their potential deviation from the core concept of our discussion section. (See the revised manuscript, discussion section)

Reviewer concern: I hope that the authors find these comments useful and are able to improve the paper to better situate such studies in the context of the literature of sexual violence in Ghana.

Author’s response: Dear reviewer, we thank you for your patience in reading and practical suggestions, allowing us to better improve this manuscript. We have included all your constructive comments and suggestions and corrected the whole document accordingly. (See the revised manuscript)

Reviewer #2

This paper add

---

## [Editor Report · Decision Letter 1]

23 Sep 2024

Factors associated with sexual violence against reproductive-age women in Ghana: a multilevel mixed-effects analysis

PONE-D-24-14631R1

Dear Dr. Mekuria Negussie,

We’re pleased to inform you that your manuscript has been judged scientifically suitable for publication and will be formally accepted for publication once it meets all outstanding technical requirements.

Kind regards,

Jessica Leight, PhD

Academic Editor

PLOS ONE
---

## [Editor Report · Acceptance letter]

24 Sep 2024

PONE-D-24-14631R1 

PLOS ONE

Dear Dr. Mekuria Negussie, 

I'm pleased to inform you that your manuscript has been deemed suitable for publication in PLOS ONE. Congratulations! Your manuscript is now being handed over to our production team.

Kind regards, 

on behalf of

Dr. Jessica Leight 

Academic Editor

PLOS ONE